# Reflections on Odor Management for Animal Feeding Operations

**Kevin Janni**

Department of Bioproducts and Biosystems Engineering, University of Minnesota, St. Paul, MN 55108, USA; kjanni@umn.edu

**Abstract:** Most animal feeding operation owners recognize that they need to manage odors from their operations as part of their social relationship with their neighbors and local community. That was not always the case. Odors, whether pleasant or unpleasant, can evoke strong emotions and physiological responses. Odors from animal feeding operations are normally considered unpleasant and offensive if strong smelling and smelled often or for long periods of time. Animal feeding operation owners need to be aware of their odor emissions and the impacts the odors have on their neighbors and community. Good neighbor relations and effective communications can help identify odor problems and communicate what is being done to manage them. Odor management research and education includes odor basics, key processes including generation, emissions and dispersion, impacts, community and neighbor relations, and numerous mitigation practices. Animal feeding operation owners considering practices to reduce odor emissions or their impacts need to weigh the costs, expected effectiveness, and how the practice fits into the overall operation. Policymakers need science-based information to make informed decisions that balance the concerns and needs of neighbors and the community and the businesswomen and men that own and operate the animal feeding operations. This paper provides a broad overview of animal feeding operation odors and odor management.

**Keywords:** odors; animal agriculture; animal feeding operation; odor sources; odor mitigation; biofilters; covers; dispersion

## 1. Introduction

Most owners and managers of animal feeding operations (AFOs) recognize that they need to manage odors from their operations as part of their social relationship with their neighbors and local community. To manage odors from AFOs effectively, it is important to understand odors and their impacts on neighbors and the community, as well as the sources and activities that emit odors, mitigation practices, siting tools, other useful resources, and the importance of good communication with neighbors about odors and the odor management practices being used.

Animal feeding operations, which are defined by the US Environmental Protection Agency as "agricultural operations where animals are kept and raised in confined situations" [1], generate a variety of emissions that affect the air quality in and around the operation. The materials emitted include odors; gases such as hydrogen sulfide ($H_2S$), ammonia ($NH_3$), and methane ($CH_4$); particulate matter; and bio aerosols. This paper is focused on odors. For this paper, odors are defined as a mixture of volatile gases and vapors that come from the animals; fresh and stored manures that include feces, urine, bedding, spilled feed and water; and processes including manure treatment and land application. Odors from AFOs can come from feed materials (e.g., silage), exhaust ventilation air from animal barns, emissions from open animal lots, manure storages and/or treatment facilities (e.g., manure storages, treatment lagoons, solid manure piles, and compost sites), poorly managed animal mortalities,

manure storage agitation, and land application. This paper provides an overview of odors and odor management concepts for AFOs. Other resources described in this paper can provide more detailed information about practical odor management practices and the research basis for this paper.

## 2. Olfaction—the Sense of Smell

Humans detect odorous compounds with their olfactory epithelium, located high in the rear of their nasal cavity [2,3]. The olfactory epithelium has receptors that characterize an odor depending on both the composition and concentrations of the numerous odorous molecules in the air near the olfactory epithelium [2,3]. People can enhance odor detection by sniffing, which improves contact between the odorous molecules in the air and the olfactory epithelium cilia [3]. People can reduce odor detection by breathing through their mouth, which reduces inhaled air contact with the olfactory epithelium.

A small percentage of people are hypersensitive to odors and are able to detect odors at concentrations much lower than the general population. Another small percentage of people are anosmic and have a very poor sense of smell. The sense of smell of people in general has a typical bell shaped distribution between hypersensitive and anosmic people [4]. This variation in people's sense of smell is one important factor to consider when deciding if and how much odor management may be needed.

Another important characteristic of the sense of smell is that continued exposure over a long period of time to an odor or inhalation of a very strong odor can cause olfactory adaptation or fatigue [4]. When a person experiences olfactory adaptation or fatigue, the person's sense of smell becomes less sensitive. Usually, a person's sense of smell returns to normal after the odor exposure is removed. Olfactory fatigue explains why some people, experiencing a new or different odor, detect and smell something while others, exposed to the odor for a long time, may not detect an odor.

## 3. Odors

Odors evoke a wide range of physiological and emotional reactions. Some odors are energizing, others are calming. Different people can have different reactions to the same odor. Bakeries, coffee shops, aromatherapy suppliers, and the perfume and cologne industries are examples of businesses and industries that use odors and scents to evoke positive physiological and emotional reactions. Pleasant food smells can make your mouth water with anticipation. Unpleasant smells, like burnt popcorn or rotting garbage, can cause you to crinkle your nose or have a gag response.

Most odors are a mixture of many different gases or vapors at extremely low concentrations. Both the gas composition and their concentrations affect the perceived odor. Some gases in the mixture can be at very low concentrations and still contribute to the perceived odor of the mixture. The very low concentrations and large number of gases that comprise an odor make it very difficult and expensive to identify and measure the concentrations of all of the gases in an air sample that contribute to the odor. Measuring some of the gases present in the air usually is not sufficient to characterize the odor of an air sample.

One way to measure and describe odors is to use olfactometry. Olfactometry uses trained individuals and standardized procedures to measure odor levels and characteristics [3,5,6]. One advantage of olfactometry is that it has a direct relation between the sample odor and the human sense of smell. The method also analyzes the entire gas mixture so that gases at very low concentrations are included in the analysis. One disadvantage with olfactometry is that the method lacks precision. Even trained individuals vary in their sense of smell [7].

## 4. Odor Characteristics

Olfactometry can be used to assess several odor characteristics, including the detection threshold, recognition threshold, intensity, hedonic tone (i.e., unpleasantness to pleasantness), and character descriptors [3]. The detection threshold of an odorous air sample is the amount of non-odorous air needed to dilute the odorous air to a concentration at which trained people can just detect that the

diluted air stream is different when compared to non-odorous air. The recognition threshold of an odorous air sample is the amount of non-odorous air needed to dilute odorous air to a concentration at which trained people can recognize the diluted air stream odor.

Odor intensity describes the strength of an odor sample when the concentration is above the recognition threshold. Table 1 lists descriptors for a 0 to 5 intensity scale [8]. Air with no detectable odor has an intensity of zero and is not annoying. Air with a very faint odor has an intensity of one and is not annoying, while air with a faint odor has an intensity of two and is considered to be a little annoying. Air with a moderate odor, strong odor, and very strong odor has intensities of three, four, and five, respectively, and increasing annoyance.

**Table 1.** Odor intensity scale [8].

| Reference Scale | Odor Intensity Strength | Odor Intensity Annoyance |
| :---: | :---: | :---: |
| 0 | No odor | Not annoying |
| 1 | Very faint | Not annoying |
| 2 | Faint | A little annoying |
| 3 | Moderate | Annoying |
| 4 | Strong | Very annoying |
| 5 | Very strong | Extremely annoying |

The hedonic tone describes the unpleasantness or pleasantness of an odor. Character descriptors are used to describe the odor using terms like minty, citrusy, earthy, or flowery [3].

When managing odors from AFOs, the important characteristics are the detection threshold and intensity. Odors from animal manures are generally considered unpleasant. When odors are very unpleasant, they commonly are described as offensive and annoying. If odors are below their detection threshold, the concentration at which no odor is detected, the odor is considered very well managed. If odors are detected, a person's and the community's reaction can depend on the frequency of odor detection, intensity when detected, detection duration, and the offensiveness of the odor detected. These four characteristics are key to odor management, which is discussed later.

## 5. Animal Feeding Operation Odors

Animal feeding operations emit hundreds of volatile compounds that contribute to the odors from the operations [9–11]. The compounds can be categorized as acids, alcohols, aldehydes, amides, amines, aromatics, esters, ethers, fixed gases, halogenated hydrocarbons, hydrocarbons, ketones, nitriles, other nitrogen-containing compounds, phenols, sulfur-containing compounds, steroids, and other compounds [9–11]. The list of compounds has increased in length with time, going from 72 compounds listed by Kreis [9], to 168 compounds in a list by O'Neill and Phillips [10], to a list of more than 400 compounds by Schiffman et al. [11]. Many of the detected compounds in air from AFOs were at very low concentrations (e.g., parts per billion (ppb) and parts per trillion (ppt)) which makes their measurement very difficult and expensive.

Ammonia and hydrogen sulfide are common gases emitted from AFOs at concentrations that can be measured in the field. While both are important components of AFO odors, many other compounds contribute to odors. Zahn et al. [12] found that olfactory responses varied as the concentrations of nine volatile organic compounds (VOCs) were changed in an 18-compound synthetic swine odor solution. These studies indicate that the VOC composition and concentrations found in air from AFOs are important elements of their odors.

## 6. Health and Nuisance Concerns

Gases and odors from AFOs can raise health and nuisance concerns. The concerns may depend on the concentration or intensity, duration, frequency, and offensiveness sensed by people. The commonly

very low concentrations, complex mixtures, and variability of odors from AFOs make assessing their impacts very difficult.

Some gases from AFOs (i.e., $H_2S$, $NH_3$ and $CH_4$) can be found, under certain circumstances, at sufficiently high concentrations to cause severe health problems and death. The most common circumstance is during manure storage agitation prior to manure pumping and land application. Poultry barns with manure storage inside the barn can also have $NH_3$ concentrations sufficiently high to be unhealthy for the birds and animal care givers in the barn [13,14]. Methane gas can accumulate in swine finishing barns without adequate ventilation to levels that can lead to explosions if an ignition spark is provided [15]. Hydrogen sulfide can be released from agitated stored manure at sufficiently high concentrations to cause people to pass out and die [16]. These examples represent conditions when gases from AFOs are dangerous. Proper manure and ventilation management can prevent these dangerous conditions.

A systematic review of research studies on the association between AFO proximity and the health of individuals living near them found no consistent dose relationship between exposure and disease [17]. A follow up review found no consistent dose-response relationship between exposure and disease except for an increased incidence of Q-fever, associated with a proximity to goat production facilities [18]. If questions about the health effects of living near AFOs need to be answered, O'Connor et al. [17] recommended large, long-term prospective studies.

Some gases emitted from AFOs are irritants [11]. Associated health complaints include eye, nose, and throat irritation; headaches; nausea; hoarseness/cough; nasal congestion; shortness of breath; stress; drowsiness; and altered mood [19]. Reactions to odorants can vary widely depending on individual tolerance to the individual and composite compounds. Schiffman et al. [19] recommended more research to determine the role of odor intensity, duration, and degree of unpleasantness on health symptoms from exposure to odorous emissions from AFOs.

Animal feeding operation odors can also lead to complaints from neighbors and community members because they detect odors they consider objectionable or offensive which interfere with their enjoyment of outdoor areas on public and private property and in their homes. Objectionable odors can negatively impact people's well-being. Reactions to odors depend on the frequency, intensity, duration, and offensiveness of the odors detected. Offensive odors at detectable levels can annoy people and make them more tense and angry. Odor complaints are normally a local issue because most odorous mixtures can be diluted to below detection levels given enough time, distance, and turbulent air mixing [20]. In some communities, odor complaints are handled as nuisances [8]. Air emissions from AFOs can raise legitimate concerns among neighbors and community members if owners and operators fail to manage odor emissions.

## 7. Acceptable Community Odor Levels and Perspectives

Acceptable odor management must consider multiple perspectives. Neighbors and local community members want sufficient odor management so they do not need to worry about or experience negative health impacts or be unable to enjoy their homes and public areas. Animal feeding operation owners and managers want odor management to be adequate to meet community expectations, economical, and to fit within their management and operating plans. Agricultural communities need to balance the concerns and needs of neighbors, the community, the environment, and the businessmen and women that own and manage AFOs.

Animal feeding operation owners and managers need to remember that odors evoke both emotional and physiological responses as they develop plans to manage odors from their operation. Some neighbors may not be bothered by some odors, while others may feel that their home and property has been invaded when they detect odors from nearby AFOs. Other community members and the public may become upset when they smell odors at their property lines; homes; and nearby public areas, including public roads, schools, parks, and towns. Neighbors and community members

need to recognize that some odors are common in and around AFOs and the cropland where manure is land applied. Communities need to come to an agreement on how much odor is acceptable.

## 8. Odor Management Goals

Odor management describes what an animal feeding operation needs to do to meet neighbor and community odor expectations with regards to odor exposure and perception. People's odor exposure and perception depends on the frequency, intensity, duration, and offensiveness of odors [8]. These terms create an acronym, FIDO [21]. Odor management can be accomplished by managing FIDO. Most agricultural communities will accept some detectable odors from AFOs. Some communities may tolerate objectionable odors occasionally. For good odor management, it is important for communities to discuss and come to agreement on what are acceptable odor conditions in terms of frequency, intensity, duration, and offensiveness.

Odor frequency describes how often a neighbor or community member detects odors from an AFO. Some communities may allow detectable odors multiple times a week or several days during certain periods of time. For example, communities may recognize that odors will be detected more frequently during the time when manure storages are agitated and the manure is removed and land applied at agronomic rates based on approved nutrient management plans. At other times of the year, neighbors and community members may expect no detectable odors for months at a time. Odors are generally more likely during mild and warm weather and when wind speeds are low [20].

Odor intensity describes the odor strength or concentration (Table 1). For good odor management, it is desirable to have downwind odor intensities that are not annoying (i.e., intensities 0 or 1) most of the time. Some communities may allow odors that might be considered a little annoying (i.e., intensity 2) some of the time. For good odor management, it is desirable to avoid odors that are recognizable and have higher odor intensities (i.e., intensities 3, 4, or 5) that may be considered annoying to extremely annoying.

Odor duration describes how long an odor is detected at different intensities. If odors are detected for only a few seconds or minutes, a person may find the odor acceptable. If odors are detected for hours, days, or longer time periods, a person may find the odor exposure unacceptable.

Odor offensiveness describes a person's reaction to the odor. Odors can range from pleasant to unpleasant. When odors are very unpleasant, they can quickly become offensive and spark strong negative emotional reactions. Odors from AFOs can be offensive if they are too intense.

Animal feeding operation owners and managers can manage odors from their operations by being mindful of their odor emissions and the impact the odors from their operation has on their neighbors and the community. They can also manage odors by managing the frequency, intensity, and duration that their operations emit odors at levels that are likely to be detectable and annoying or objectionable downwind. Once owners and managers recognize that some of their odor emissions may be detectable, they can investigate practices that will mitigate those emissions and reduce the odor intensity, duration, and offensiveness of the odor emitted from their operation. The goal of odor management is to reduce odor concentrations from levels where neighbors and the public can detect them to non-offensive levels most of the time.

Odor management is different than the management of regulated air pollutants or toxins. Odor management is different because many of the gases and vapors that comprise an odor are at extremely low concentrations with no known negative impacts on the downwind environment or public health when below detectable levels.

## 9. Good Neighbor Relations and Communications

Good neighbor relations and communications can help animal feeding operation (AFO) owners be aware of potential and existing odor issues. Effective communications can be especially important when planning to build a new or expand an existing AFO. Discussions with neighbors early in the planning process can be an opportunity to hear about and maybe alleviate existing and/or potential

odor concerns from proposed new or expanded operations. Being open to neighbor input about odor issues and letting neighbors know that owners and managers are concerned about odors can help alleviate odor concerns. It is also important for owners and managers to communicate what is being done to monitor and mitigate odors. Without effective communications, neighbors that detect odors may assume that the owner does not care about odors and is not doing anything to monitor and manage odor emissions. It is commonly recommended that owners and managers visit neighbors at least once or twice a year to discuss odors and give updates on odor management plans and activities. Owners and managers are also encouraged to have established procedures for receiving and addressing odor complaints.

## 10. Siting Animal Feeding Operations

One important consideration to better manage odors from AFOs is to site the AFO, manure storage and treatment areas, and fields for land application away from neighbors, public parks and lakes, schools, religious institutions, towns, and cities. Remote sites give emitted odors time and distance to disperse to non-detectable or barely-detectable levels. Some states and local units of government specify separation distances between AFOs, neighbors and towns [8].

Some research groups in Midwest states have developed site assessment tools that people planning to build new or expand existing AFOs can use to assess the potential odor impact the proposed operation might be expected to have on neighbors. The tools consider feedlot size, production practices, and local weather conditions that impact odor dispersion. If the tools indicate that odor might be a problem, the owner can consider implementing mitigation practices or consider alternative locations. Most tools are based on allowing some detectable odors. Actual weather conditions may result in more or fewer odor events and more or less intense odors. A link to a video about these tools for estimating setbacks is available online on the air quality page of the Livestock and Poultry Environmental Learning Community (LPELC) website [22].

## 11. Finding Odor Problems

The first step towards addressing an odor problem is to determine if and when an AFO has odor problems. A good practice is to visit the neighbors to the operation and fields on which manure is land applied every six to twelve months. Valuable information can also be gained from employees, farm visitors, and consultants. If odors are suspected, a non-smoking person with a good sense of smell can be asked to drive around the operation before visiting to sniff the air downwind and record whether or not they smelled farm odors. Sampling needs to be done before visiting the site and being exposed to the odors to avoid olfactory adaptation or fatigue. If odors are detected downwind of the operation multiple times in a week, it is likely that the operation has an odor problem.

## 12. Prioritizing, Implementing, and Assessing Odor Management Practices

Animal feeding operation owners and managers who decide that they want or need to implement one or more odor mitigation practices need to identify all odor sources and odor-emitting practices. Many odor sources were listed in the introduction. Small sources that emit highly offensive odors may be important sources too. Wet and moldy feed or poorly managed dead animal compost piles can be examples. Once odor sources or activities are identified, they can be prioritized to identify which source or activity needs to be managed differently to reduce odor emissions. Information from neighbors may help set priorities.

Once a source or activity has been selected to be managed differently, potential mitigation practices need to be investigated. Owners and managers need to consider the advantages and disadvantages of each practice, the costs, and how the practices will fit into the overall operation and management plan. After considering options, a practice should be implemented. After a practice has been implemented, the effectiveness of the practice should be assessed to determine whether or not the practice is (1) reducing odor levels, (2) fitting into the operation, and (3) being economical. It is good practice to

visit neighbors and others impacted by the operation's odors as part of the assessment to tell them about what is being done to manage odors and get feedback on the effectiveness of the practices used.

This process of identifying sources or activities that may need odor mitigation, investigating odor mitigation practices, and implementing and assessing practices may need to be repeated several times to achieve adequate odor management to meet the needs of the owner and the community. In many cases, AFOs that treat air from the most odorous sources may obtain sufficient emission reduction to manage nearby odor levels.

## 13. Odor Mitigation Practices

Odor mitigation practices continue to be researched and developed. Practices used need to be effective, economical, and fit into the operation. One or more practices may be needed for an AFO to achieve the level of odor management needed to meet community and owner expectations.

Animal feeding operation owners and managers that have determined that they have an odor issue and want to implement one or more odor management practices will want to investigate existing practices. There are reviews that summarize published research on odor mitigation practices [8,23,24]. There are also websites that provide practical information about odor management practices [22,25]. The LPELC website has videos, factsheets and archived webinars that provide practical, science-based information about practices that reduce airborne emissions [22]. The Air Management Practices Assessment Tool (AMPAT) website, developed by Iowa State University faculty, provides an overview of mitigation practices for emissions from animal housing, manure storage and treatment, and land application [25]. The AMPAT site also has an extensive literature database [26].

The three research reviews describe numerous technologies to reduce odor emissions from AFOs [8,23,24]. Sweeten et al. [8] summarized technologies for sources, including animal buildings, open lots, manure treatment systems, storages, land application, and handling animal mortalities. Rahman and Borhan [23] and Liu et al. [24] reviewed practices and technologies for odor control in swine facilities. The types of technologies were similar to those reviewed by Sweeten et al. [8].

General descriptions of some common odor mitigation practices available are summarized here. Readers interested in more detailed information about the technologies summarized here are encouraged to review the three reviews [8,23,24], the AMPAT [25,26] databases, and the original research publications.

Table 2 summarizes the effectiveness and cost of some of the odor control technologies reviewed by Liu et al. [24]. The AMPAT website also provides tables summarizing effectiveness and estimated costs [25].

**Table 2.** Summary of odor control technologies in swine facilities [24].

|  | Technology | Effectiveness | Overall Cost |
|---|---|---|---|
| Ration/diet modification | Low crude protein diets and/or feed additives | Moderate | Low |
| Manure handling and treatment | Solid-liquid separation | Moderate | Moderate to high |
|  | Storage additives | Uncertain | Moderate |
|  | Impermeable storage covers | High | Moderate |
|  | Permeable storage covers | Moderate | Low to moderate |
|  | Anaerobic digestion | High | High |
| Air treatment | Oil spraying | Low to moderate | Moderate |
|  | Biofilters | High | Low to moderate |
|  | Wet scrubbers | Moderate | Moderate to high |
|  | Vegetative environmental buffers | Low to moderate | Low |

### 13.1. Animal Diet/Ration Management

The concept behind diet/ration management for odor management is precision feeding, which formulates rations that provide all of the essential nutrients and energy needed for growth while



minimizing the unused nutrients in voided feces and urine, enteric emissions, and subsequent emissions from stored and land-applied manure [8,23,24]. One approach reduces the crude protein levels in the feed while adding amino acids needed for proper growth and animal well-being. With less excess protein, there will be less nitrogen and sulfur in the manure and subsequently less $NH_3$ and $H_2S$ given off from the manure. Phase and split-sex feeding can be used in swine and poultry production. Another approach uses feed additives or better feed processing techniques to improve nutrient absorption and reduce unused nutrients in the manure. Carter et al. [27] summarized common diet /ration management practices for odor management.

### 13.2. Dust Management With Oil Sprinkling and Fat Addition

Odorous gases can adhere to dust particles and be detected later when the gases desorb. Substantial amounts of odorous compounds and $NH_3$ emitted from swine buildings are adsorbed and transported by dust particles [28,29]. One method for reducing dust emissions and the odorous gases adhering to the dust is to sprinkle small amounts (i.e., 2 ounces per 100 square feet) of biodegradable vegetable oil in swine facilities [30,31]. A Midwest Plan Service factsheet by Zhang [32] describes the method. Another method for reducing airborne feed particulate matter is to add fat or oil to the feed [33–36].

### 13.3. Manure Handling and Treatment

Manure handling and treatment practices used between when feces and urine are voided and the manure is land applied can impact odor emissions. Microorganisms in manure produce odorous compounds as they biodegrade the manure if they have the appropriate nutrients, moisture, time, and environmental conditions for growth. The frequent collection and removal of manure, urine, or wet bedding reduces odor emissions if the manure is either land applied or placed in a storage facility that limits emissions.

The following sections outline some common manure handling and treatment systems for odor management. Some technologies are specific to specific manure management systems and AFOs. Some technologies are not good options for some types of operations. More information about these treatments can be found in the reviews by Sweeten et al. [8], Rahman and Borhan [23], and Liu et al. [24]. General information on manure treatment is also available on the LPELC Air Quality and AMPAT websites [22,25].

#### 13.3.1. Solid/Liquid Separation

Solid/liquid animal manure separation can be used in manure handling and treatment systems that handle each fraction (i.e., solid and liquid) separately. Solid/liquid separation can involve mechanical separation only or mechanical separation plus chemical treatment. Chemical treatment is used to cause coagulation and flocculation to remove smaller solids from the liquid fraction. Some dairy operations use mechanical separation to obtain solids for use as bedding or soil amendment. The liquid fraction can be stored before it is land applied. The liquid portion of the separated manure and recycled flush water can be very odorous. Zhu et al. [37] reported that removing total suspended solids with only mechanical separation might not be enough to reduce the odor generation from pig slurry; chemical treatment may be needed. Several studies have used chemicals to enhance solid/liquid separation with conventional separators [38,39].

#### 13.3.2. Drying Manure Solids

Drying solid manure can reduce odor emissions during subsequent storage. Manure with 25% or more solids can be handled as a solid. If dried to 50% or more solids, the manure normally will be sufficiently porous to permit enough air diffusion to avoid anaerobic decomposition [40]. Odor management may be required during the manure drying process to avoid unacceptable odor emissions. Energy costs make manure drying an expensive odor management option.

### 13.3.3. Aerobic Manure Treatment

Aerobic manure treatment systems maintain aerobic conditions (with oxygen) in collected and stored manure, so that manure constituents are broken down by aerobic microorganisms into more stable materials that allow storage with little odor [41,42].

Aerobic treatment systems used by AFOs include aerobic lagoons, aerated lagoons, surface aerated lagoon or storage, and solids composting [42–44]. All aerobic treatment systems need to be designed, operated and managed properly to effectively and economically manage odors [45]. Aerobic lagoons are a treatment system that can be used in regions where warm water temperatures promote year-round biodegradation. Excessive volatile solids loading, the accumulation of volatile solids during cold weather, reduced microbial activity, or inadequate aeration can lead to incomplete degradation and the production of odorous compounds [42]. Mechanical aeration can be used to reduce the lagoon surface area needed for air exchange. Surface aeration attempts to reduce the energy needs for aeration [46,47]

Composting is a well-established and documented aerobic treatment system for stabilizing solid manure and animal mortalities. Composting systems need proper operation and management to avoid offensive odors. The right carbon to nitrogen ratio, moisture, and adequate aeration are needed for effective composting without generating offensive odors. Practical information about composting livestock and poultry manures [43] and composting livestock and poultry moralities [44] are available on the LPELC webpages.

### 13.3.4. Anaerobic Manure Treatment

Anaerobic manure treatment systems maintain anaerobic conditions (without oxygen) in the collected and stored manure so that the manure constituents are broken down by anaerobic microorganisms into more stable materials that allow storage with little odor. Intermediate products during anaerobic breakdown can be volatile and odorous. Products and intermediates from anaerobic decomposition are generally considered more odorous compared to aerobic treatment.

Anaerobic manure treatment systems include anaerobic lagoons and anaerobic digesters. Anaerobic lagoons need to be designed, operated, and managed properly to avoid generating offensive odors [42,45,48]. Anaerobic digesters can be designed to generate $CH_4$ gas that can be used to generate heat or electrical power. The capital and operating costs of anaerobic digesters have limited their use in odor mitigation [23,24].

### 13.4. Manure and Litter Additives

Chemical and biological additives can be used to reduce odor generation or emissions during manure transport, storage, agitation, and land application. Chemicals can be added to oxidize odorous volatile compounds, adjust pH, or react with volatile compounds and form precipitates. Biological additives attempt to change the biochemical pathways that produce odorous gases. Reviews by McCrory and Hobbs [49], Sweeten et al. [8], Rahman and Borhan [23], and Liu et al. [24] include information on numerous manure additives. Most additive research studies have mixed results. While individual farm results vary, additives are worth considering if more odor control is needed. Two factsheets about additives for swine manure and poultry litter were written by Shah et al. [50,51] and are available online.

### 13.5. Manure Storage Covers

Covers on manure storages reduce odor emissions by creating a barrier between the stored manure containing odorous gases and the airflow above it [52]. Covers can be either permeable or impermeable. Permeable covers include the natural crusts that form on dairy manure storages that use organic bedding; straw blown onto swine manure storages [53]; and synthetic materials, including clay balls and geotextile membranes [54]. Permeable covers allow gas molecules and water to pass through, but odorous gas emissions are reduced compared to uncovered manure. Impermeable covers

include plastic, rubberized, or concrete covers that trap most gas molecules between the cover and the manure [55]. Impermeable covers can be inflated or have a vacuum drawn on them to reduce the impact of wind on the cover. Plastic covers can be expensive to install but they can cut emissions by nearly 100%. Plastic covers also require extra equipment and management to deal with the gases generated under the covers and the bubbles they can produce, precipitation collected on the top and wildlife or loose farm animals that try to walk across a plastic cover. Impermeable covers can also complicate manure agitation and removal for land application.

## 13.6. Biofilters

A biofilter is a biologically based treatment that uses microbes on a solid media to treat barn exhaust ventilation air or air from manure storages to reduce odor emissions [56–59]. Biofilters have a porous media that supports a biofilm in which microorganisms absorb and break down odorous gases and vapors from the air blown through the media. The microorganisms use the energy and nutrients in the gases and vapors to grow and reproduce. Well designed and managed biofilters can reduce odors and $H_2S$ by as much as 95% and $NH_3$ by 80% [60,61]. Biofilters require moisture management to sustain microbial growth to achieve effective odor reduction and can be built in different configurations [62–64]. Media selection is an important design consideration [60,61,65,66] to maintain microbial populations and porosity to avoid excess pressure drops. Janni et al. [67] provide summary information about biofilters.

## 13.7. Wet Scrubbers

Wet scrubbers include a variety of air cleaning devices that use a liquid, usually water, to remove dust and soluble gases and vapors from the air. They can be called particulate scrubbers, spray towers, or packed gas absorption towers. If acid is added to the water, packed towers can be very effective at removing $NH_3$, with removal efficiencies of between 90% and 99% when the scrubbing solution had a pH of 4 or lower [68]. Prototype spray towers had $NH_3$ removal efficiencies of between 27% and 77%, depending on the inlet $NH_3$ concentration, number of stages, and air velocity [69]. Melese and Ognik [68] reported that the average odor removal efficiencies of the two packed towers tested were 29% and 34%, respectively. The variation in the odor removal was high, with a minimum removal efficiency of 3% and a maximum of 51% [68]. Manuzun et al. [70] provide an overview of wet scrubbers for mechanically ventilated animal facilities.

## 13.8. Enhanced Dispersion

Enhanced dispersion dilutes odorous gases in the air to concentrations below detection levels. Odor dispersion can be enhanced using vegetative environmental buffers and trees; wind walls; chimneys; and increased separation distances between the odor sources and neighbors, towns, schools, parks, and other public areas [71–73]. Trees and wind breaks may improve the perceived appearance of an AFO by hiding odor sources in addition to enhancing dispersion. Tyndall and Colletti [74] reviewed research on shelterbelts for odor management around swine facilities.

## 13.9. Land Application

Stored manure agitation and land application are common manure handling practices that emit strong odors. Direct manure injection or incorporation soon after surface application can reduce odor emissions during or after land application [75] and subsequent detection downwind. Avoiding practices that spray odorous manure into the air can reduce emissions. Applying manure on sunny and windy days will enhance the odor dispersion. The wind direction impacts which nearby neighbors and public areas may detect odors during and after land application. Checking with neighbors prior to manure agitation and land application in order to avoid performing agitation and land application just prior to or during special neighbor or community events (i.e., holidays, parties, or family get-togethers) would help to maintain good neighbor relations. There are stories of some AFO owners paying

for hotel stays for close neighbors who would be impacted during manure storage agitation and land applications.

## 14. Assessing Alternative Management Practices

Many AFO owners and managers find it difficult to assess their operations and investigate the impact of different practices on airborne emissions. A group of agricultural partners and universities developed an interactive online tool for owners and managers to assess their operations. The National Air Quality Site Assessment Tool (NAQSAT) is a confidential and free web-based tool that helps assess the impact of different management practices on airborne emissions from AFOs [76]. NAQSAT has species-specific questions for dairy, beef, swine, broiler chicken, laying hen, turkey, and horse operations. There are questions on animals, housing, diet, manure management, land application, mortalities, neighbor relations, and gravel road management that help owners and managers evaluate practices and potential options to mitigate odors.

## 15. Conclusions

Animal feeding operations and some of their manure handling practices (such as land application) can be odor sources that nearby neighbors and community members may find objectionable. Odors from AFOs commonly include hundreds of different gases and particulate matter that contribute to the odor intensity and hedonic tone (i.e., pleasantness vs. unpleasantness). Owners and managers of AFOs need to recognize that odors can generate strong physiological and emotional reactions. Odors can be nuisances that disturb/disrupt people's ability to enjoy their homes and public areas. Odors may irritate peoples' eyes and throats and alter a person's mood, but a research review did not find a consistent dose relationship between gas exposures and disease [17,18]. To manage AFO odors, owners and managers are encouraged to manage the frequency at which odors are detected, the intensity of the odors detected, the duration that the odors are detected, and the odor offensiveness. By managing the odor FIDO (i.e., frequency, intensity, duration, and offensiveness), the odor concerns of neighbors and community members can be managed. Good neighbor relations and communications about odors and the odor management practices being used can also help to manage community odor concerns. If additional odor management is needed to meet community expectations, it is recommended that owners and managers identify key sources, prioritize sources needing mitigation, investigate possible mitigation practices, implement one or more practices, and assess the impact of the mitigation practices adopted. Prescriptive odor management practices are not recommended because AFOs are highly variable and the amount of mitigation needed can vary. Regular communications with neighbors can help owners and managers to learn about odor concerns and give them a chance to tell neighbors about what is being done to manage odors from their AFO. There are online resources that can help AFO owners and managers to find research-based odor management tools and resources.

**Funding:** This research received no external funding.

**Acknowledgments:** The author acknowledges input and comments by Erin Cortus and knowledge gained by collaborating with colleagues, including Larry Jacobson, Chuck Clanton, Richard Nicolai, Neslihan Akdeniz, Jun Zhu, Huiqing Guo, Jose Bicudo, David Schmidt, Brian Hetchler, and graduate and undergraduate students.

**Conflicts of Interest:** The author declares no conflict of interest.

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
