# Peer review of "Reflections on Odor Management for Animal Feeding Operations"

_atmosphere, doi:10.3390/atmos11050453_

Round 1
Reviewer 1 Report
The manuscript thoroughly reviews AFO odor issues, odor management goals, and odor control practices. The revision is well written and organized.
Line 20: Policy makers should be Policymakers.
Author Response
The single change recommended for line 20 was made.
Reviewer 2 Report
Overview & General Recommendations
This review manuscript holistically discusses general concepts related to odor originated from animal feeding operations. Well-rounded insights are provided on a broad range of aspects that are supported by thoroughly reviewed literature and augmented with a synthesis of the topics. Because of this, this review covers a topic of relevance and general interest to the readers of this journal.
The manuscript reads satisfactorily, with very few technical and non-technical English language and style errors throughout. More commas could be added. The manuscript is logically organized and flows clearly between sections. Each section is further well-organized, covers topics in-depth, and synthesizes the content such that a general conclusion and future direction can be discerned. Overall, a well-written, clear, and concise review.
Major comments
- Compared to other reviews on this subject, what makes this review, unique and novel? No doubt the literature is thoroughly reviewed and analyzed; but how does this review advance the current state of knowledge on this topic? There are several resources and manuscripts on related topics, so a few statements that address the aforementioned questions would be well-received. The inclusion of community and neighbor relations is often not discussed, this might be a topic to highlight.
- It may be worth considering an additional section on improving implementation of odor mitigation practices. There are a lot of resources and tools available for owners and managers of AFOs, but implementation is somewhat intermittent, what can be done to spur the usage of odor mitigation practices? Is it a lack of technical knowledge in the research or maybe, the capability to design, install, and operate? The author’s perspective on this would interesting.
- Also, it may be worth considering an additional section on the research of odor mitigation practices. What do researchers need to consider when designing a new technology, during field validation/testing, or what are some best practices, measurements, etc. that need to be consider? Similarly, the author’s perspective on this would could help shape the robustness and direction future odor mitigation research studies.
- There are several statements that might require citations, if they are available.
Minor comments
Consider using abbreviations in general, but mainly for ‘animal feeding operations’, AFOs
Consistency is needed on: i.e., that is; e.g., for example; etc. with punctuation and usage
Consider adding this additional reference for related to AMPAT: Maurer, D. L., Koziel, J. A., Harmon, J. D., Hoff, S. J., Rieck-Hinz, A. M., & Andersen, D. S. (2016). Summary of performance data for technologies to control gaseous, odor, and particulate emissions from livestock operations: Air management practices assessment tool (AMPAT). Data in brief, 7, 1413-1429
L31 add comma after effectively
L31-34 sentence is potentially too long, while clear, it takes careful reading to follow
L37 is ‘materials’ the best word?
L39 add comma after ‘paper’
L47 change ‘report’ to ‘paper’
L55 add .. which reduces ‘inhaled’ air contact…
L69-81 are there references for these statements?
L101 add comma after ‘five’
L121 lowercase ‘p’ in parts
L133 phrase seems awkward
L135-144 reference for any of these statements?
L161 is ‘and their homes’ considered private property?
L166 add comma after ‘communities’
L193 add comma after ‘management’
L233 change ‘chance’ to ‘ opportunity’
L263 add comma after ‘suspected’
L270 make manager plural
L272 remove double period
L277 add comma after ‘differently’
L296-L305 maybe provide direct hyperlink in text to these websites
L306 incomplete sentence, please revise
L340 not sure if I follow the meaning of this sentence
L354 add comma after i.e.
L380 change ‘mechanically’ to ‘mechanical’
L460 add comma after ‘vary’
L421 what’s the difference between wildlife and animals?
L424 add ‘exhaust’ after ‘barn’
L439 change ‘of’ to ‘or’
L466 add ‘operations’ after ‘owners’
Author Response
Comments related to major changes
- The manuscript is more of comprehensive overview rather than a literature review. The overview, as based on the original proposal, is based on my experience. The overview provides a holistic way to address odor issues. As noted by the reviewer, community and neighbor relations is not often considered and in my opinion it is very important.
- The last paragraph of Section 12 is one way to continuously improve odor management practices. “This process of identifying sources or activities that may need odor mitigation, investigating odor mitigation practices, implementing and assessing practices may need to be repeated several times to achieve adequate odor management to meet the needs of the owner and the community.” If the animal feeding operation owner makes odor management a priority, does the monitory and community relations, they will learn what it takes to make their implementation adequate for the their needs.
- I would draw the reviewer’s attention to the second paragraph in Section 12. The paragraph is written for animal feeding operation owners but it also tells researchers the criteria that owners are considering – “the practice is 1) reducing odor levels, 2) fitting into the operation and 3) being economical”.
- Six reference were added including the one recommended by the reviewer.
Comments related to the minor changes:
An acronym was used for animal feeding operation (AFO) was added to the text except for the abstract and when starting a sentence. H2S, NH3 and CH4 were used too.
The use of ex. was eliminated. Use of i.e. and e.g. was reviewed.
The recommended reference was added.
The remaining recommended changes were made except for the following:
L 31-34. The sentence is essentially a list of factors that need to be considered.
L 37 Materials was retained because it is a broader term because at this point I want to start with a long list of airborne things that are emitted from animal feeding operations before focusing on odors.
L69-81 References were not added. This is general knowledge gained over the years from my perspective.
L 133. The sentence was not changed. I prefer to make the point that odors are highly variable and are made up of complex mixtures of gases at low concentrations before saying that they are difficult to assess.
L296-L305 – Hyperlinks is up to the publisher. Let me know if they are wanted.
L306 No problem identified with sentence.
L340 Manure handling and treatment begins once the feces and urine are voided and continues until the manure is land applied. How that is done impacts the odor emissions.
Round 2
Reviewer 2 Report
The author has adequately addressed the feedback - well done.
This manuscript is a resubmission of an earlier submission. The following is a list of the peer review reports and author responses from that submission.
Round 1
Reviewer 1 Report
The paper is well written and organized. It summarizes the odor issues associated with animal feeding operations and management practices for odor management. The information in the paper is important for the public, academia, government to understand the complexity of odor issues and provide essential knowledge on techniques for odor mitigations.
Specific commons:
L11: whether pleasant "or" unpleasant.
L135: The danger of ammonia and methane were discussed. Please also mention the danger of H2S that threatens animals and humans during pump-out and agitation.
L163-165: Odor complaints are normally a local issue because most odorous 163 mixtures can be diluted to below detection levels given enough time, distance and turbulent air 164 mixing. Please provide a reference and evidence here.
How about the health concern of odor as a precursor for PM or ozone?
L200: Odors are generally more likely detectable during mild and warm weather and when wind 200 speeds are low. Please provide a reference for this statement.
Author Response
See attached pdf.

Reviewer 2 Report
The Paper "Reflections on Odor Management for Animal Feeding Operations" is written in a good English language and style. It is clearly legible. It gives an overall view of odor guidance and opportunities how to proceed in case of odor complaints. It gives qualitative information.
The text should be shortened. In some chapters the text is too textbook oriented. General information to odor is too broadly presented. There is to be shortened, to focus on odor of animal feeding operations.
The structure could be improved. To that, a proposal is given in detail.
In some chapters (1, 8, 9, 10), citing of literature could be added, as it was done in ch. 7 and 14.
More comments in detail:
ch = chapter, L = line
L2: The title could be altered, instead of “Odor Management”, e.g. “Odor Guidance” or “Odor Orientation Guide”.
Abstract:
L9:…that they need to mitigate odors from…
L11: …pleasant or unpleasant…
ch. 1: Introduction:
The term "animal feeding operation" should be defined. The ch. 1 could be combined with ch. 5 and 6, directly focused on odor from animal feeding operations.
ch. 2, 3 and 4:
These chapters are on a very general level, not specified to the topic and title. Repetitions appear with ch. 9.
ch. 2, 3 and 4 could even be omitted or at least compressed and combined.
ch. 4:
L85-91: Olfactometry ist only one method to quantify odor. Field plume inspections, questionnairies etc. could be added. Or the methods completey deleted.
There is a overlapping with ch. 9. A combination is recommended.
L98, table 1:
The gradation of odor intensity strength and of odor intensity annoyance need not necessarily be as shown. It could be scrutinised.
“From the perception of an unpleasant odour to the making of a complaint, the process is characterised by repeated confrontation and the inescapability of the situation for the affected persons, as well as adverse effects on well-being and quality of life (Van Harreveld, 2001).”
This citation shows that odor annoyance is a more complex development and not only resulting from odor intensity strength.
L102:
Animal feeding operation odors can be pleasant, e.g. hay; there are not only unpleasant ones.
ch. 9:
L 185: Heading “Odor characteristics” or “Factors for assessing odor impact” instead of “Odor management”
“Odor management” should show how odours are being managed and controlled so as to prevent or minimise the release of odours from the site.
ch. 12:
“A non-smoking person … before starting to work..”
An employee is not independent from the farm. The time of checking the odour situation should be aimed at the observed situations with problems.
ch. 13:
L292: “…may need to be repeated…” Sentence to be completed.
ch. 14:
The mitigation practices are listed qualitatively. The results in the original papers are not always so clear, e.g.: Most studies with animal diet show no significant effect with regard to odor (in contrast to ammonia).
Can some of the wording be clarified?
L445: Melse and Ogink: Spelling.
A few mitigation practices, which are so far not so promising could be omitted.
ch. 15:
The Tool seems to be a first qualitative step. Measuring the effect of a mitigation needs a more targeted approach. The situation is made more difficult by the fact that other influences can also be relevant with a before and after comparison.
ch. 16:
L493: Only “good relations and communications” is probably not sufficient to achieve odor reduction. It can help to analyse the situation and to avoid escalation.
Author Response
See attached pdf.
